# An economic evaluation of an online computer-tailored smoking cessation intervention that includes message frame-tailoring: A randomized controlled trial

**Maria B. Altendorf** [1] *, **Julia C. M. van Weert**[2], **Ciska Hoving** [3], **Eline S. Smit**[2]

**1** Institute of Medical Sociology and Rehabilitation Science, Charité –Universitätsmedizin Berlin, corporate member of Freie Universität Berlin, Humboldt-Universität zu Berlin, and Berlin Institute of Health, Germany, **2** Department of Communication Science/Amsterdam School of Communication Research (ASCoR), University of Amsterdam, Amsterdam, Netherlands, **3** Department of Health Promotion/Care and Public Health Research Institute (CAPHRI), Maastricht University, Maastricht, Netherlands

* m.altendorf@googlemail.com

**Data Availability Statement:** The data file can be found at OSF: https://osf.io/4fh6g/files/osfstorage/62d123a8588bb92068b86e6d.

## Abstract

Evidence of economic evaluations of behaviour change interventions is scarce, but needed to guide policy makers' decision-making. This study economically evaluated 4 versions of an innovative online computer-tailored smoking cessation intervention. The economic evaluation from a societal perspective was embedded in a randomized controlled trial among 532 smokers using a 2 (message frame-tailoring, i.e. *how* messages are presented: autonomy-supportive vs controlling) x 2 (content-tailoring, i.e. *what* content is presented: tailored vs. generic) design. Both kinds of tailoring, content-tailoring and message frame-tailoring, were based on a set of questions asked at baseline. Self-reported costs, prolonged smoking abstinence (cost-effectiveness) and quality of life (cost-utility) were assessed during a 6-month-follow-up. For cost-effectiveness analysis, costs per abstinent smoker were calculated. For cost-utility analysis, costs per QALY (i.e. quality-adjusted life year) gained were calculated. A willingness-to-pay (WTP) threshold of €20.000 was used. Bootstrapping and sensitivity analysis were conducted. Cost-effectiveness analysis showed that up to a WTP of €2.000, the combination of message frame- and content-tailoring dominated all study groups. From a WTP of €2.005, the content-tailored group dominated all study groups. Cost-utility analysis revealed that the combination of message frame-tailoring and content-tailoring had the highest probability of being the most efficient study group at all levels of the WTP. The combination of message frame-tailoring and content-tailoring in online smoking cessation programmes seemed to have high potential for cost-effectiveness (smoking abstinence) and cost-utility (quality of life), thus providing good value for money. Yet, when the WTP for each abstinent smoker is high (i.e., €2.005 or higher), the addition of message frame-tailoring might not be worth the effort and content-tailoring only is preferred.

**Funding:** This work was supported by the Dutch Cancer Society (KWF) under Grant KWF 2015-7913.The funders had no role in study design, data collection and analysis, decision to publish, or preparation of the manuscript.

**Competing interests:** The authors have declared that no competing interests exist.

## Introduction

Smoking tobacco is the single most preventable cause of non-communicable diseases, such as cancer [1,2,3]. In 2017, about 9% of the disease burden in the Netherlands was attributable to smoking [4]. Besides a risk for individual and population health, smoking-related illness and premature death also leads to decreased labour productivity and increased health care costs for the society as a whole [1,2]. Behavioural support through online *computer-tailored* smoking cessation interventions has been shown to be an effective measure in improving quit rates among smokers over and above more static interventions, such as generic online smoking cessation information [5,6]; however, effect sizes remain small [6]. Therefore, a recent randomized controlled trial (RCT) was set out to test the (cost-)effectiveness of message frame-tailoring [7]–a novel technique of tailoring, adjusting ***how*** information is presented to people based on their individual characteristics. That is done, for instance, by adding words that are meant to communicate decisional freedom for people with a high need for autonomy or by taking a more directive communication style for people with a lower need for autonomy–in an online smoking cessation intervention, in addition to content-tailoring, i.e., adjusting ***what*** information is communicated based on people's individual situation, behaviour and beliefs. Thus, in the tailored study arms, participants received smoking cessation advice based on their answers to certain questions at baseline, such as an assessment of their level of addiction and their attitudes regarding quitting smoking as input for content-tailoring and an assessment of their need for autonomy as input for message frame-tailoring [7,8,9] Evidence of economic evaluations of online behaviour change interventions is scarce, but needed to guide policy makers' decision-making [10]. Economic evaluations measure two parameters: costs and effects, but can also determine the value for money in terms of effects. Not necessarily, the cheapest treatment or intervention option lead to the best health outcome or quality of life. Thus, the current analysis sought to answer whether the novel online smoking cessation intervention with the combination of message frame-tailoring and content-tailoring is more cost-effective and has higher cost-utility compared to message-frame-tailoring only, content-tailoring only and/or no tailoring at all, over a follow-up period over six months.

## Methods

### Study design and randomization

The present study was conducted as part of a randomized controlled trial (RCT), which aimed to test the effectiveness, cost-effectiveness and cost-utility of message frame-tailoring in a web-based computer-tailored smoking cessation intervention. Full details of the study methods and effectiveness results have been described elsewhere [7,8,9]. As depicted in Fig 1, at baseline, participants were randomized into one of four study groups: 1) message frame-tailoring (i.e. tailoring *how* messages are communicated) combined with content-tailoring (FCT); 2) message frame-tailoring only (FT); 3) content-tailoring only (CT); 4) and no tailoring at all (i.e., control group). Randomization took place at the individual level and was done through computerized randomization software.

### Intervention

In the content-tailored study groups, participants received messages that were adapted according to their responses to the baseline questionnaire. In addition, participants' need for autonomy was assessed in the baseline questionnaire to inform message frame-tailoring. Both the questionnaire and tailored message content were based on a previously developed (cost-)effective intervention [11–13] and theoretically grounded in the I-Change Model [14] while

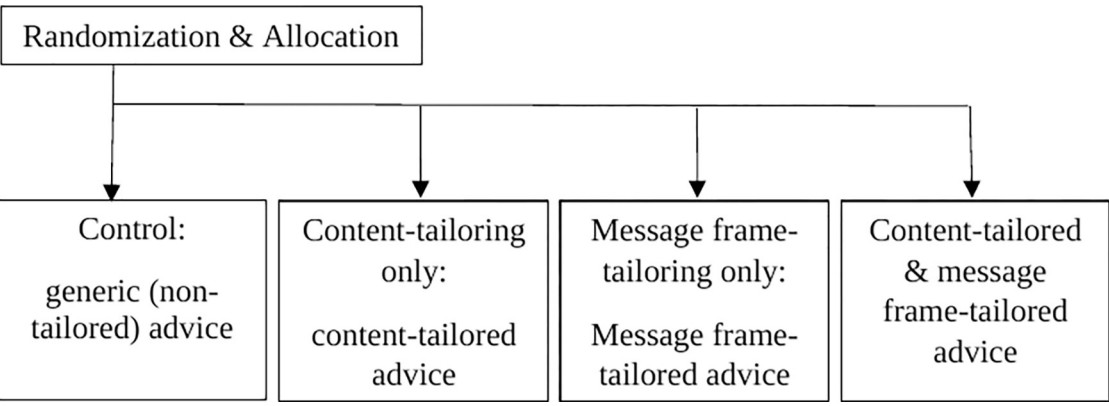

**Fig 1. Illustration of study arms.**

(questions required for) message frame-tailoring was grounded in Self-Determination Theory (SDT) [15]. At baseline (T0), participants provided information on their smoking behaviour (e.g. cigarettes smoked per day), sociodemographic (e.g. age, gender, living arrangement, educational level) and medical status (e.g. presence of respiratory or cardiovascular diseases and–in case of female gender–current pregnancy), I-Change constructs (e.g. attitude, intention to quit smoking), and the SDT-derived construct need for autonomy. When randomized into one of the three experimental study groups, at baseline (see Fig 1), participants in the content-tailoring groups received tailored advice at several times throughout the intervention, which took participants approximately 20 minutes to complete. The flow of participants is illustrated in Fig 2. In the message frame-tailoring groups, participants received this advice either in an autonomy-supportive (e.g. minimization of pressure on the reader and the possibility to choose a quit smoking date) or a controlling message frame (e.g. directive and rather forceful language and a given quit smoking date), depending on their need for autonomy. Participants in the control group completed the same questionnaire, but only received generic smoking cessation information. Immediately post-intervention (T1), which is at the same day of the intervention, all participants answered questions about perceived message relevance, health care costs, and their quality of life. One-month post intervention (T2), effectiveness measures were assessed (for more detail, see description measures below). During the six-months follow-up measurement (T3), all participants received another survey with–next to effectiveness measures–questions about their health care costs and quality of life.

## Recruitment and participants

During the recruitment phase (December 2018 until April 2019), participants were recruited via paid Google and Facebook advertisements, unpaid social media posts, online newspaper articles and radio station announcements. Eligible participants were Dutch adult smokers of 18 years or older, who had access to the internet, were proficient in Dutch and intended to quit smoking within the next six months. This RCT was approved by the Institutional Review Board (reference number 2017-PC-7599) and is registered with the Dutch Trial Register (NL6512/NRT-6700).

## Measures

Next to the variables that were measured at T0 to develop the tailored messages (see Intervention description), immediately post-intervention (T1) we assessed self-reported costs for the

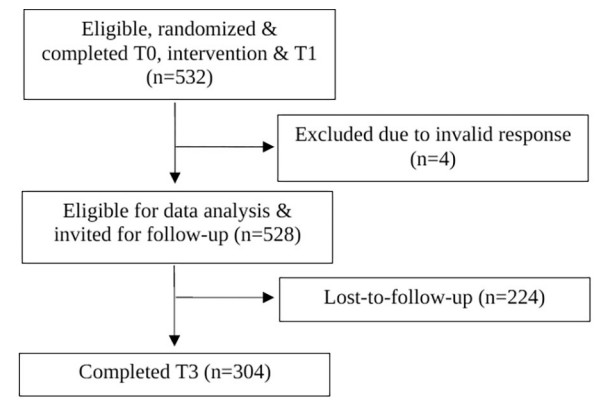

T0 (baseline pre-intervention):
Enrolment, allocation & baseline measurements for content-tailoring & message frame-tailoring

PAS online intervention: Assessment & smoking cessation advice with a duration of approx. 20 minutes

T1 (baseline immediate post-intervention):
Measurement of participant costs & QALYs

T2 (one-month post-intervention):
Measurement of psychological variables & smoking cessation for effectiveness evaluation

T3 (6 months post-intervention):
Measurement of smoking abstinence, participant costs & QALYs for cost-effectiveness evaluation

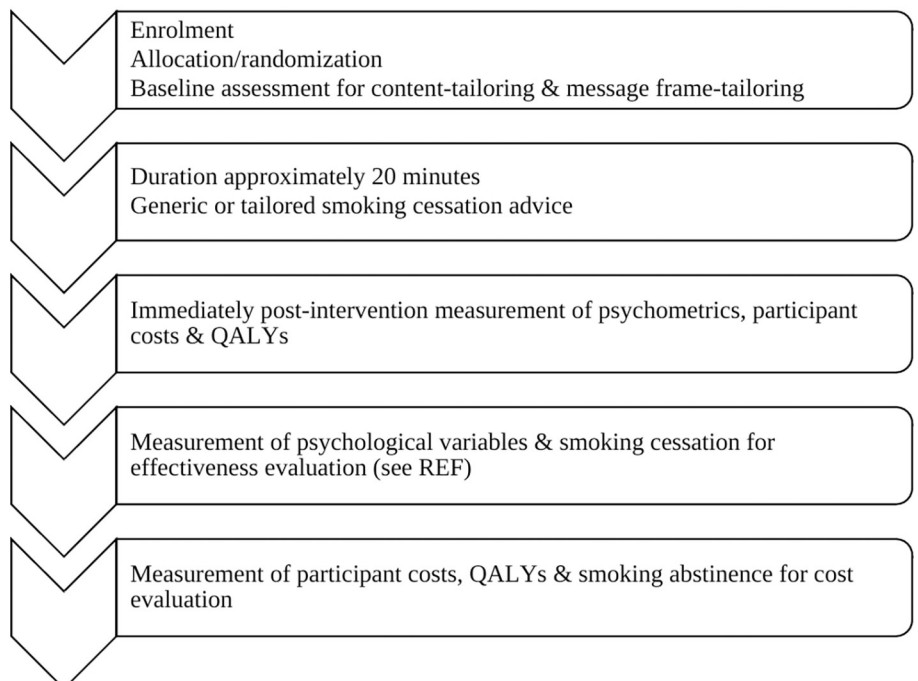

Enrolment
Allocation/randomization
Baseline assessment for content-tailoring & message frame-tailoring

Duration approximately 20 minutes
Generic or tailored smoking cessation advice

Immediately post-intervention measurement of psychometrics, participant costs & QALYs

Measurement of psychological variables & smoking cessation for effectiveness evaluation (see REF)

Measurement of participant costs, QALYs & smoking abstinence for cost evaluation

**Fig 2. Flow chart & timeline of participants in the RCT.**

cost-effectiveness analysis (CEA; in addition to smoking-related behaviour) and for the cost-utility analysis (CUA, in addition to quality of life). At six-month follow-up (T3), these measures were assessed again.

**Intervention costs.** Intervention costs consisted of hosting costs for the website in which the tailoring software was integrated and hosting costs for the tailoring software itself. We did neither consider costs for the development of the intervention nor research-specific costs, as these were not necessary for intervention implementation. Intervention costs were the same for all three experimental study groups (i.e., FCT, FC, CT), namely 0.41 euro per participant.

For the control group, costs were lower (i.e., 0.19 euro per participant) as for this group only the website hosting costs had to be considered as they received a generic advice for which no tailoring software was necessary.

**Health care costs.** Health care costs were assessed with a six-month retrospective online questionnaire with open-ended questions, which was based on a previously developed questionnaire [15]. Participants reported the number of contacts with a general practitioner, practice assistant, (practice) nurse, mental health specialist, inpatient and/or outpatient specialist, occupational physician, lifestyle coach, social worker, pharmacist (assistant), and alternative medicine professional. Also, participants reported the number and duration of hospital admissions, and the frequency and duration of other care (e.g., ambulant care, informal care, rehabilitation, dental care). Moreover, we assessed information about the dose, duration and frequency of prescribed smoking cessation medication (e.g. Varenicline, Bupropion, Nortriptyline) and other medication, as well as over-the-counter nicotine replacement therapy (e.g. nicotine patch or nicotine gum). Then, when necessary, i.e. when participants reported cost prices from other years than 2018 (such as from 2019, which was when the trial was finished), cost prices were indexed to the year 2018 by using the consumer price index of 103.44 [16]. All costs were then summed up which resulted in total costs per participant.

**Smoking abstinence.** The outcome measure for the CEA was seven-day point prevalence abstinence from smoking. This was measured by asking participants whether they had smoked in the last seven days (yes = 0, no = 1).

**Quality of life.** The outcome measure for the CUA was quality of life, assessed in terms of quality-adjusted life years (QALYs). We used two different validated self-report measures to calculate utility scores and have conducted the cost-utility analyses with both measures separately: 1) the ICECAP-A, and 2) the Euro-QoL (EQ-5D-5L). The *Dutch ICECAP-A* [17,18] assessed participant's quality of life as their general capability in life regarding five attributes: attachment, stability, autonomy, achievement, and enjoyment (1 = not capable; 2 = a little capable; 3 = quite capable; 4 = fully capable). ICECAP-A scores per attribute were translated into tariff scores [17,18], resulting in an overall, accumulated utility value for capability, ranging from -0.024 (no capability) to 1 (full capability). The *Dutch EQ-5D-5L* [19] comprised five rather physical dimensions: mobility, self-care, usual activities, pain/discomfort and anxiety (1 = no problems; 2 = slight problems; 3 = moderate problems; 4 = severe problems; 5 = and extreme problems). QALYs from the ICECAP-A measure were computed by taking the average of ICECAP-A utility scores from the baseline and follow-up measurements. QALYs assessed with the EuroQoL were assessed by taking the average EQ-5D-5L utility scores from the baseline and the follow-up measurement. Resulting QALYs scores thus represented the number of QALYs lost or gained during the six-month study period. For instance, gaining 0.5 QALYs can be understood as gaining 0.5 years in full capability or health, or gaining 1 year in less than perfect capability or health.

## Economic evaluation & data analysis

**Economic evaluation approach.** This economic evaluation consisted of a CEA and a CUA. Unlike CEAs that compare the effectiveness outcome (here smoking cessation) of the intervention as comparator, CUAs use an increase or decrease in utilities, such as QoL, as the outcome for analysis [20,21]. For the economic evaluation, four steps were taken [20–23]. First, we determined that a societal perspective would be used to also consider costs relevant for society, such as intervention costs, costs for medication and alternative health care providers, and participant costs. Second, relevant health care (-related) costs and effects were identified. Third, costs and effects were compared by calculating the net monetary benefit (NMB) as

four study groups had to be compared. The NMB was calculated for each intervention group in relation to a reference group. The NMB is the difference in effects between two groups multiplied by a cost-effectiveness or cost-utility threshold (i.e. society's willingness to pay (WTP) per abstinent patient or per QALYs gained), subtracted with the difference in costs between these groups, according to the below formula:

$$\text{NMB} = (\text{effects}_i - \text{effects}_c) \text{ x WTP} - (\text{costs}_i - \text{costs}_c)$$

In our analyses, a WTP of €20.000 per QALY was used as this is an accepted Dutch threshold for health promotion interventions [22]. The intervention can be considered cost-effective or has a high cost-utility, respectively, when the NMB is positive–thus, has a value of 0 or above. In a fourth step, uncertainty and sensitivity analyses were conducted to test the robustness of the results.

In all analyses, cost-effectiveness and cost-utility acceptability curves (all abbreviated CEACs) were plotted based on the outcomes from 1000 bootstrap iterations. The CEAC illustrates the probability of an intervention to be cost-effective compared to a reference group (i.e. in our case the control group) considering the WTP and taking modelling uncertainty into account [22].

Calculations of the NMB, CEAC, and bootstrap analyses were conducted using Microsoft Office Excel 2016. Other analyses were conducted using the statistical computer software IBM SPSS 25.0.

## Data analysis approach

The primary and main analysis was conducted with complete cases only, i.e. only with participants who had complete data for both the baseline and the six-month follow-up measurement, and taking into account intervention costs. Also, two sensitivity analyses were conducted to deal with uncertainty and demonstrate robustness of our results. A first sensitivity analysis (called first sensitivity analysis hereafter) was conducted with complete cases, not taking intervention costs into account. The second sensitivity analysis (called second sensitivity analysis hereafter) was performed including cases with missing values on the six-month follow-up measurement, with missing values being imputed using the last observation carried forward (LOCF) method and taking into account the intervention costs.

- In the main analysis, we analysed complete cases and did consider intervention costs;

- In the first sensitivity analysis, we analysed complete cases without intervention costs being considered;

- In the second sensitivity analysis, we analysed all cases applying the LOCF method to impute missing data points, while again considering the intervention costs.

The LOCF method imputes missing data based on available data from the previous assessment (i.e. baseline assessment). In line with the Russell Standard, in this second sensitivity analysis missing data for smoking abstinence were replaced according to a conservative "negative" scenario, that is, participants lost to follow-up were considered to still be smoking [23,24].

## Results

### Sample characteristics and participant flow throughout the RCT

As illustrated in Fig 1, at baseline, 528 participants were eligible for analysis, of which 304 participants (57.6%) were followed-up after six months and 224 participants were lost-to-follow-

up (42.4%). No significant baseline differences were found between participants who did and who did not complete the six-month follow-up questionnaire regarding demographics, chronic diseases and smoking-related behaviors.

## Baseline characteristics

Table 1 provides an overview of the sample characteristics per study group at baseline. Comparison of participants in the different groups with regard to their demographics (i.e. age, sex, living arrangement and educational level), smoking-related behaviours and chronic conditions did not show significant differences between study groups.

However, at baseline, significant mean differences were found between participants' total health care-related costs (F = 4.235 (3,1), p = .006). Tukey's b post-hoc test showed that the FCT and control group significantly differed, as well as the CT and the control group (see Table 1). In addition, significant baseline differences were found between the FCT group and the control group in QALYs assessed with the ICECAP-A tool (F = 2.686 (3,1), p = .046); participants in the FCT group had a significantly higher QoL compared to those in the control condition. At six-month follow-up, these significant differences were not present anymore.

As shown in Table 2, smoking abstinence, i.e., the likelihood of people being abstinent from smoking for the last 7-days (at 6-months follow-up) was highest in the CT group, before and after bootstrapping. The cost-utility outcome, quality of life, was highest for people in the FCT group, when considering the ICECAP tool or in the FT group when QoL was measured with the EuroQoL tool.

## Economic evaluation

**Cost-effectiveness analysis.** As shown in Table 2, both before and after bootstrapping, the costs in the FCT group were lowest among all groups. Moreover, as illustrated in Fig 2, at the commonly used €20.000 Dutch cut-off value for the cost-effectiveness of preventive interventions, NMB analyses revealed that the CT group had an 85% likelihood to be most cost-effective intervention (striped upper line). Thus, the CT group seemed to be the most cost-effective group when smoking abstinence at six-month follow-up was considered as the outcome measure for analysis. Yet, it is worth mentioning that up to a WTP of €2.000, the FCT study group dominated all other study groups, whereas from a WTP of €2.005 and over, the likelihood that the CT study group was the most cost-effective increased.

Results from the first sensitivity analysis confirmed that, this time with a probability of 83%, the CT group was likely to be the most cost-effective, something which was also confirmed in the second sensitivity analysis (i.e. probability cost-effective of 82% at WTP of €20.000) (see S1 Table).

**Cost-utility analysis.** As presented in Table 2, health effects in terms of QALYs were highest in the FCT group. In Fig 3, the results from the NMB analyses with the ICECAP-A and the EuroQoL utility score, respectively, as dependent variable are presented. Both CUAs illustrate that at the commonly accepted WTP of €20.000 for preventive interventions, the FCT group had the highest probability to be the most cost-effective, i.e. 60% for ICECAP and 85% for EuroQoL, respectively. A similar pattern was found in the results from the first and second sensitivity analysis (see S1 Table).

## Discussion

This is the first study, as far as we aware, that aimed to economically evaluate whether an online smoking cessation intervention that combined message frame-tailoring and content-tailoring is more cost-effective and has higher cost-utility compared to an intervention with

**Table 1. Demographics, background variables, mean health care costs and effects per study group at baseline.**

| | FCT (n = 135) | | FT (n = 123) | | CT (n = 144) | | Control (n = 126) | |
|---|---|---|---|---|---|---|---|---|
| | *N* | *%* | *n* | *%* | *n* | *%* | *n* | *%* |
| **Demographics** | | | | | | | | |
| Sex (male) | 45 | 24.10 | 47 | 25.10 | 51 | 27.30 | 44 | 23.50 |
| Age (years), M (SD) | 41.99 (14.13) | | 41.50 (13.19) | | 41.91 (14.84) | | 41.07 (13.61) | |
| **Educational level** | | | | | | | | |
| High | 61 | 45.20 | 56 | 45.50 | 63 | 43.80 | 53 | 42.10 |
| Middle | 60 | 44.40 | 58 | 47.20 | 60 | 41.70 | 50 | 39.70 |
| Low | 14 | 10.40 | 9 | 7.30 | 21 | 14.60 | 23 | 18.30 |
| **Living arrangement** | | | | | | | | |
| With partner | 36 | 26.70 | 19 | 15.40 | 33 | 22.90 | 22 | 17.50 |
| With partner and child(ren) | 30 | 22.20 | 29 | 23.60 | 31 | 21.50 | 29 | 23.00 |
| With child(ren) | 8 | 5.90 | 16 | 13.00 | 15 | 10.40 | 16 | 12.70 |
| Alone | 50 | 37.00 | 50 | 40.70 | 59 | 41.00 | 49 | 38.90 |
| Other/missing | 11 | 8.10 | 9 | 7.30 | 6 | 4.20 | 10 | 7.90 |
| **Number of daily smoked:** | | | | | | | | |
| Cigarettes, M (SD) | 11.72 (7.92) | | 11.31 (8.52) | | 10.49 (8.07) | | 11.01 (8.93) | |
| Shags, M (SD) | 2.63 (6.78) | | 3.69 (7.76) | | 4.24(8.63) | | 4.21 (9.61) | |
| Cigars, M (SD) | 0.10 (0.96) | | 0.71 (4.25) | | 0.44 (2.54) | | 0.38 (2.02) | |
| Cigarillos, M (SD) | 0.04 (0.44) | | 0.32 (1.84) | | 0.23 (1.50) | | 0.15 (1.36) | |
| Pipes, M (SD) | 0 (0) | | 0.20 (1.53) | | 0.28 (1.74) | | 0.02 (0.13) | |
| Earlier quit attempts, M (SD) | 5.23 (12.57) | | 4.24 (6.19) | | 6.77 (17.89) | | 6.59 (13.46) | |
| Age of starting to smoke, M (SD) | 18.76 (6.05) | | 18.07 (4.49) | | 17.46 (4.98) | | 16.63 (4.33) | |
| **Existence of (chronic) disease [a]:** | | | | | | | | |
| Heart disease | 8 | 5.90 | 8 | 6.50 | 12 | 8.30 | 11 | 8.70 |
| COPD | 20 | 14.80 | 22 | 17.90 | 36 | 25.00 | 28 | 22.20 |
| Diabetes | 4 | 3.00 | 5 | 4.10 | 8 | 5.60 | 8 | 6.30 |
| Cancer | 3 | 2.20 | 8 | 6.50 | 6 | 4.20 | 12 | 9.50 |
| **QALY:** | | | | | | | | |
| EQ-5D-5L, M (SD) | -.27 (.23) | | -.23 (.28) | | -.226 (.26) | | -.211 (.29) | |
| ICECAP-A, M (SD) | .86 (.15)[c] | | .83 (.18) | | .81 (.18) | | .79 (.19)[c] | |
| **Health care-related costs [b]:** | | | | | | | | |
| Health care costs, M (SD) | 320.05 (668.77) | | 453.88 (712.66) | | 335.76 (621.73) | | 611.47 (1390.42) | |
| Hospital, M (SD) | 372.18 (2223.34) | | 362.59 (1127.09) | | 192.10 (671.52) | | 712.41 (2865.49) | |
| Medication, M (SD) | 22.59 (113.17) | | 78.77 (287.84) | | 47.95 (225.69) | | 149.09 (738.68) | |
| Other care, M (SD) | 128.95 (563.16) | | 381.74 (1557.13) | | 157.59 (592.32) | | 401.76 (1628.86) | |
| Total costs per participant [f], M (SD) | 844.21 (2642.99)[d,] | | 1277.41 (2395.96) | | 733.82 (1349.38)[e] | | 1874.92 (4440.96)[d, e] | |

*Note.* FCT = Frame-tailoring & Content-tailoring. FT = Frame-tailoring. CT = Content-tailoring.

[a] = Percentages don't add up to 100%, as only those participants with the medical condition are presented in the table.

[b] = Costs were assessed in relation to the prior 6 months.

[c] = Significant mean difference $M_{FCT}$ = .86 vs. $M_{control}$ = .79.

[d] = Significant mean difference $M_{FCT}$ = 844.21 vs. $M_{control}$ = 1874.92.

[e] = Significant mean difference $M_{CT}$ = 733.82 vs. $M_{control}$ = 1874.92.

[f] = Total cost calculations might differ slightly from total health care costs calculated in SPSS, due to decimal rounding–the SPSS-calculated were used for the analyses and are therefore presented in the last row.

**Table 2. Mean costs & effects per study group at 6-month follow-up (primary analysis).**

| Type of analysis | Mean costs in € | | | | Mean effects [a] | | | | Probability of highest NMB [b], % | | | |
|---|---|---|---|---|---|---|---|---|---|---|---|---|
| | FCT | FT | CT | Control | FCT | FT | CT | Control | FCT | FT | CT | Control |
| Trial data (Before 1000 bootstrap iterations) | | | | | | | | | | | | |
| Abstinence | 612 | 1854 | 802 | 1308 | 0.29 | 0.29 | 0.38 | 0.19 | | | | |
| QoL (EQ-5D-5L) | 612 | 1854 | 802 | 1308 | -0.18 | -0.18 | -0.24 | -0.25 | | | | |
| QoL (ICECAP-A) | 612 | 1854 | 802 | 1308 | 0.85 | 0.85 | 0.85 | 0.82 | | | | |
| After 1000 bootstrap iterations | | | | | | | | | Primary analysis | | | |
| Abstinence | 472 | 2593 | 692 | 1596 | 0.31 | 0.31 | 0.38 | 0.18 | 11 | 4 | 85 | 0 |
| QoL (EQ-5D-5L) | 593 | 2677 | 724 | 1303 | -0.21 | -0.18 | -0.21 | -0.29 | 85 | 11 | 4 | 1 |
| QoL (ICECAP-A) | 429 | 1887 | 860 | 1085 | 0.88 | 0.82 | 0.85 | 0.85 | 60 | 4 | 34 | 3 |

*Note.* FCT = Frame-tailoring & Content-tailoring. FT = Frame-tailoring. CT = Content-tailoring. EQ-5D-5L = the 5-level EQ-5D version by the EuroQoL group to assess health-related quality adjusted life years in economic evaluations. ICECAP-A = ICEpop CAPability measure for Adults, a broader measure for health-related quality adjusted life years in economic evaluations. NMB = Net monetary benefit.

[a] = Mean effects for smoking abstinence = average likelihood to be abstinent from smoking for the last 7-days ranging from 0 (no) to 1 (yes).

[b] = at a WTP threshold of 20.000 euro per health unit increase.

message-frame-tailoring only, content-tailoring only and/or no tailoring at all. In this study, intervention effects were assessed in terms of smoking abstinence (i.e. for cost-effectiveness) as well as quality of life (i.e. for cost-utility) at six-month follow-up, with costs assessed during the six-month follow-up period.

## Cost-effectiveness of message frame-tailoring

The results of the cost-effectiveness analysis revealed that up to a WTP of €2.000, the FCT study group dominated all other study groups. Yet, when the WTP would exceed €2.005 per abstinent smoker–which is not unlikely given the commonly accepted WTP-threshold of €20.000 –the content-tailoring study group had the highest probability of being cost-effective, yet costs were lowest in the FCT study group. An earlier economic evaluation of a similar online computer-tailored smoking cessation intervention also showed that online content-tailoring was most cost-effective study group when compared to care as usual (i.e. standard smoking cessation guidance by practice nurses) or the combination of practice nurse counselling with online content-tailored smoking cessation advice–also considering a WTP of €20.000 [12]. To this point of the discussion, in addition, findings from a study that investigated solely the effectiveness of this present intervention after one-month follow-up, the content-tailoring only study group seemed to increase the likelihood of smokers being abstinent, whereas the addition of message frame-tailoring seemed unable to further increase smoking abstinence [7]. Thus, to this point of the discussion, it seems that content-tailoring has great potential for (cost-) effectiveness of online smoking cessation interventions.

## Cost-utility of message frame-tailoring

In terms of cost-utility, the results give an indication–as anticipated—that the novel combination of message frame-tailoring and content-tailoring led to the highest cost-utility, thus, generating the highest QoL in the most efficient way. All sensitivity analyses showed our findings to be robust. A likely explanation for these findings as related to the cost-utility (outcome: quality of life) of the intervention, could be that a message, frame-tailored to one's personal need for autonomy, in addition to content-tailored advice based on other smoking-related and

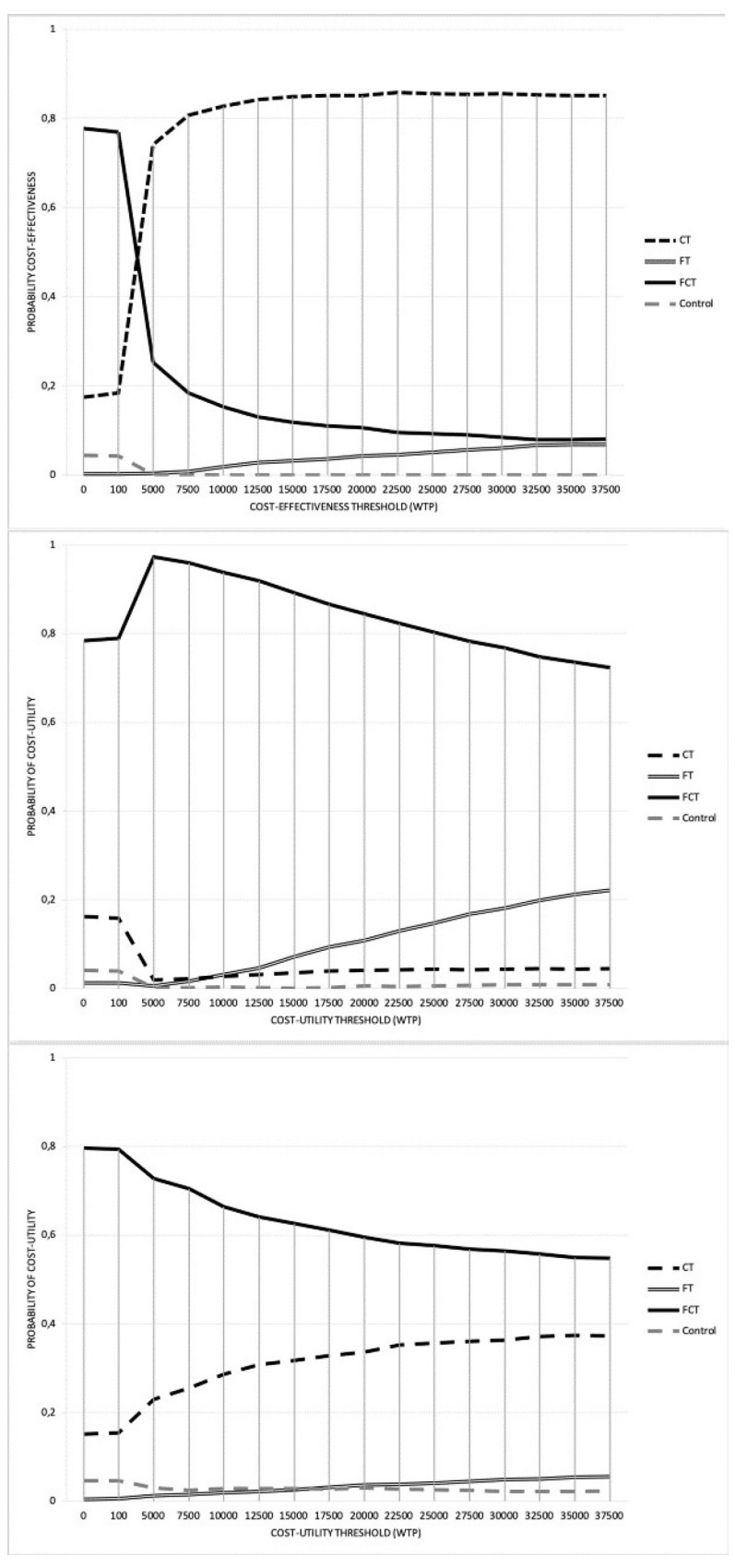

**Fig 3. Cost-effectiveness & cost-utility acceptability curves (CEACs) with 7-day point prevalence abstinence, quality of life measured with the ICECAP, and quality of life measured with the EuroQoL assessed at six-month follow-up as outcomes (primary analysis).**

cognitive parameters might be more empowering when it comes to decision-making regarding smoking cessation than a message that is not tailored in terms of its framing [26,27]. Consequently, one could assume that the message frame-tailored messages may have empowered participants to feel more capable of daily-life-activities, and subsequently score higher on related QoL assessments. For instance, the EQ-5D-5L scale [19] considered the dimension of self-care in daily activities and the ICECAP scale [17,18] assessed QoL with participant's capability of autonomy in daily life activities, which was also reflected in message frame-tailoring (i.e. messages were tailored to smoker's need for autonomy). Further research including a qualitative investigation, such as interviews with participating smokers and experts in the area of QoL assessment and emotions, might be used to investigate whether increased feelings of empowerment indeed represent a process underlying the effects of the combination of message frame- and content-tailoring on smokers' QoL.

## Suggestions for future research

Cost-utility results seemed to indicate the favourability of combining content-tailoring with message frame-tailoring under varying levels of the WTP. However, for cost-effectiveness (i.e., taking smoking abstinence as an outcome) this combination was only preferred over content-tailoring under relatively low WTP-thresholds (up to a WTP of €2000). Following, this does suggest some avenues for future research. One question that arises is whether it could be that frame-tailored messages may be able to fulfil participants' affective needs better than non-frame-tailored messages, thereby indirectly enhancing participant's QoL but not necessarily leading to more behaviour change or smoking cessation? Is seems conceivable that the frame-tailored messages might have led to more emotional need fulfilment than non-frame-tailored messages, as research on gain and loss-framing in health, advertisement and political domains showed associations between message framing and emotions [28–30]. For instance, advantage framed and gain framed messages induced positively valenced emotions (i.e. hope) [29,30]. Such emotional need fulfilment could have been captured in the QoL-measurement tools [1719], yet is likely not reflected in measures of smoking abstinence. However, to validate these assumptions–and next to the qualitative research suggested earlier–future research efforts might include sophisticated physiological investigations next to the more traditional surveys [31], such as assessment of sweat on the skin or cortisol in the salvia, to gain better insights in the positive and negative affect experienced by intervention participants (due to feeling angry/not having the amount of autonomy needed in certain situations).

## Strengths and limitations

The present study has some strengths and limitations. The first strength of the study is that we used two different assessment tools for QoL–the ICECAP-A [17,18] and the EuroQoL [19]. By conducting the cost-utility analysis independently with each QoL-questionnaire, we were able to paint a clearer picture of the cost-effectiveness of the intervention in relation to respondents' QoL. This, as the EuroQoL tool mainly assesses health-related QoL, whereas the ICECAP tool has been shown to complement the EuroQoL tool with information about one's capability of well-being [32,33]. Second, we used the bootstrapping method to minimize uncertainty in results and also performed two sensitivity analyses (with one sensitivity analysis with data from complete cases without intervention costs and one sensitivity analysis with missing data

imputed using the LOCF method and including intervention costs. Both sensitivity analyses revealed similar results and thus underline the robustness of our findings. A third strength of the present RCT is that its external validity is high, as a representative sample of smokers was recruited through real-life settings, such as social media. A potential limitation is that data used were collected using self-report measures, which could have been subject to recall bias and social desirability bias as participants might not accurately have responded in all cases as the follow-up period was six months and respondents may have responded in ways that seem to be socially more desirable [34]. To overcome such potential bias in future studies, secondary health care use data from health insurance companies and physiologically measured data of respondents emotional reactions to message frames could be used for comparison and data triangulation.

## Conclusion

The present study assessed the cost-effectiveness and cost-utility of message frame-tailoring as integrated in a content-tailored online smoking cessation intervention, compared to message frame-tailoring-only, content-tailoring-only, and generic smoking cessation information. This economic evaluation can provide a financial argument for the investment in this intervention as a nationwide, cost-effective smoking cessation programme and thereby bridge the gap between research and practice. Both cost-utility analyses (i.e., taking quality of life as an outcome with two different quality of life scales) indicated the favourability of the novel combination of message frame-tailoring and content-tailoring. The integration of the two types of tailoring can thus be considered as providing good value for money. The cost-effectiveness analysis (i.e., taking smoking abstinence as an outcome) indicated the favourability of the novel combination of message frame-tailoring and content-tailoring only up to a WTP of €2000. When the WTP for each additional abstinent smoker is €2005 or higher, the addition of message frame-tailoring might not be worth the effort and content-tailoring only is preferred.

## Supporting information

**S1 Table. Results from economic evaluation analyses based on 1000 bootstrap iterations.** (DOCX)

## Acknowledgments

The authors wish to thank Inge van Strien-Knippenberg and Dennis de Ruijter for their valuable support in the preparation for data collection. Moreover, we are very thankful to Fairouz Kasri, Silvia Evers and Thea van Asselt for their support and expertise during the data analysis.

## Author Contributions

**Conceptualization:** Maria B. Altendorf, Eline S. Smit.

**Formal analysis:** Maria B. Altendorf.

**Funding acquisition:** Julia C. M. van Weert, Ciska Hoving, Eline S. Smit.

**Methodology:** Maria B. Altendorf, Eline S. Smit.

**Project administration:** Maria B. Altendorf, Eline S. Smit.

**Resources:** Maria B. Altendorf.

**Supervision:** Julia C. M. van Weert, Ciska Hoving, Eline S. Smit.

**Visualization:** Maria B. Altendorf.

**Writing – original draft:** Maria B. Altendorf.

**Writing – review & editing:** Maria B. Altendorf, Julia C. M. van Weert, Ciska Hoving, Eline S. Smit.

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
