## [Decision Letter · Decision Letter 0]

20 May 2022

PDIG-D-21-00116

An economic evaluation of an online computer-tailored smoking cessation intervention that includes message frame-tailoring: A randomized controlled trial

PLOS Digital Health

Dear Dr. Altendorf,

Thank you for submitting your manuscript to PLOS Digital Health. Your manuscript was sent to review and it was reviewed by members of the editorial board. 

After careful consideration, we feel that it has merit but does not fully meet PLOS Digital Health's publication criteria as it currently stands. Therefore, we invite you to submit a revised version of the manuscript that addresses the points raised during the review process. In particular, Reviewer 1 provides useful constructive comments on how to improve the paper and how to provide greater clarity on your study. 

We look forward to receiving your revised manuscript.

Kind regards,

Valentina Lichtner

Academic Editor

PLOS Digital Health

Journal Requirements:

State what role the funders took in the study. If the funders had no role in your study, please state: “The funders had no role in study design, data collection and analysis, decision to publish, or preparation of the manuscript.”

2. Please update your Competing Interests statement. If you have no competing interests to declare, please state: “The authors have declared that no competing interests exist.”

3. In the online submission form, you indicated that “The data that support the findings of this study are available from the corresponding author, upon reasonable request.”. All PLOS journals now require all data underlying the findings described in their manuscript to be freely available to other researchers, either 1. In a public repository, 2. Within the manuscript itself, or 3. Uploaded as supplementary information.

4. Please provide separate figure files in .tif or .eps format only and remove any ensure that all files are under our size limit of 10MB.

For more information about how to convert your figure files please see our guidelines: https://journals.plos.org/digitalhealth/s/figures

Additional Editor Comments (if provided):

Reviewers' comments:

Reviewer's Responses to Questions

**Comments to the Author**

1. Does this manuscript meet PLOS Digital Health’s publication criteria? Is the manuscript technically sound, and do the data support the conclusions? The manuscript must describe methodologically and ethically rigorous research with conclusions that are appropriately drawn based on the data presented.

Reviewer #1: Yes

Reviewer #3: No

2. Has the statistical analysis been performed appropriately and rigorously?

Reviewer #1: N/A

Reviewer #3: No

3. Have the authors made all data underlying the findings in their manuscript fully available (please refer to the Data Availability Statement at the start of the manuscript PDF file)?

Reviewer #1: No

Reviewer #3: No

4. Is the manuscript presented in an intelligible fashion and written in standard English?

Reviewer #1: No

Reviewer #3: No

5. Review Comments to the Author

Reviewer #1: Thank you for giving me the opportunity to review this paper. This economic evaluation looks at using tailored online computer smoking cessation interventions. In a pandemic when face to face meetings are difficult and restrictions can change without warning, evaluating an online intervention is particularly timely. The evaluation includes a cost-effectiveness analysis and cost-utility analysis (the latter using 2 quality of life questionnaires).

I enjoyed reading this paper and thought the results and discussion were interesting, particularly with 2 different quality of life outcomes – both measuring different aspects of quality of life – this gave some insight into the mechanism behind the success of the different interventions and what might be driving the results.

Overall I thought the paper was good, however there were aspects of it which I struggled to understand, and I think these areas need clarification, the main issues I had was with understanding the interventions and timelines.

I have included my comments below:

I found parts of the Abstract is quite hard to follow, particularly the final paragraph of results, it took me several reads to understand. I think it would help the reader to understand this better by replacing ‘until a WTP of €2.000’ with ‘up to a WTP of €2.000’ – in the Abstract and throughout.

I was also not sure what the intervention entailed – the sentence ‘using a 2 (message frame-tailoring, i.e. how messages are presented: present vs absent) x 2 (content-tailoring, i.e. what content is presented: present vs. absent) design’ for example was hard to decipher. On first read through a short sentence explaining that the tailoring of the intervention is based on questions asked at baseline would also help. It wasn’t until I finished reading the manuscript that I understood this part of the Abstract. 

Introduction

I struggled to understand what the intervention consisted of, particularly the difference between message frame-tailoring and content-tailoring. I understand that there is a word limit but would it be possible to make the intervention clearer, maybe by clarifying the examples? 

Methods

The first sentence of the Methods section suggests that the aim of the RCT is different to the aim of the economic evaluation – the economic evaluation seeks to establish the cost-effectiveness of the message frame-tailoring and content-tailoring compared to the other 3 arms, however the RCT aim is to establish (cost-)effectiveness of message frame-tailoring. I don’t see this is a problem with the manuscript if the aims were slightly different, but it stood out.

The intervention was a bit clearer in the Methods section but with 4 arms I still struggled to conceptualise the different arms on the first reading – would it be possible to add a table or figure so the reader can refer to something visual? 

It wasn’t clear how long the intervention lasted for – maybe a visual timeline would help make this clearer? Perhaps showing how long the intervention lasted for and then include T1, T2 and T3 and what was collected at each of these timepoints (ie quality of life etc). For example, it is not clear here that quality of life is collected at baseline.

Did the baseline questionnaire inform the content-tailored messages only? The first sentence suggests this, however the next sentence described the theory behind the message frame-tailoring intervention. This just needs a bit of clarification to help the reader understand how the messages are informed.

I may have missed it, but were ethics approval details included?

In terms of collecting resource use I was unsure of the timings of these – I read that from T1 the previous 6-months resource use was collected, then at 6 months too – is this 6 months post randomisation or 6 months post intervention. Again a timeline would help with understanding this, and could reduce the word count – you could combine the timeline of the intervention, T1, T2, T3 with resource use.

Intervention costs – just out of interest were there any maintenance costs for the website or software?

The healthcare costs section is clear, however I was expecting to read about which societal non-health care resources were used as these were mentioned but not detailed. Were e-cigarette resource use included?

The sentence ‘QALYs were computed by taking either the average of ICECAP-A utility scores from the baseline and follow-up measurements, or the average EQ-5D-5L utility scores’ I found this sentence confusing – it suggests that QALYs are an average of either ICECAP-A or EQ-5Q-5L across the timepoints when these questionnaires were collected – rather than a more typical mean QALY for each arm at each timepoint and then the gain/loss calculated from that.

Please could you add an explanation/justification of why you omitted the intervention costs in both of the sensitivity analyses? Also, it would be good to see a sensitivity analysis using the Russell Standard using complete cases and including intervention costs – so it could be compared to the primary analysis.

In the section ‘Economic evaluation approach’ if you have enough word count, it would be helpful to explain to the reader that if the NMB is 0 or above the intervention (i) would be considered cost-effective.

Results

In your first paragraph, the last sentence says there was not a difference in baseline characteristics for demographics, chronic disease or smoking-related behaviours - were there any differences in other baseline characteristics (other than the QALYs and baseline costs that you mention in the next paragraph)? Or do all the baseline characteristics fall into these three categories?

Would it be possible to add a short section (one or two sentences) to summarise the main RCT results so the reader knows which intervention was more effective? I know that you have included the primary outcome of abstinence in Table 2, but a short summary earlier on will help set the context.

Were the differences in costs/QALYS adjusted for any baseline characteristics – for example baseline costs or QALYs, especially as there are imbalances in baseline costs and QALYs.

I was confused as to why the costs for the bootstrapped results were so different for each outcome? Did you use different co-variates in a regression for each? If not then maybe running more bootstraps (5000?) would result in more similar, and potentially more robust, results.

Discussion

I wasn’t sure what ‘A likely explanation for these findings as related to the cost-utility of the intervention, could be that…’ meant on pages 15-16, lines 304 to 305. Should this be ‘as related to the cost-effectiveness’?

The first sentence in ‘Suggestions for future research’ initially suggests that for both cost-effectiveness and cost-utility FCT is favoured, however, it is only at a low WTP threshold, could the sentence be re-written so that it is clear early on in the sentence that it is only at a low WTP that FCT is favoured? At a typical WTP of 20,000 Euros CT is favoured. This is repeated in the Conclusion, whilst technically correct it is a bit misleading.

In the ‘strengths’ section it says that intervention costs were included in the sensitivity analyses, however in the Methods section ‘Data analysis approach’ you state that intervention costs were excluded – this just needs amending to be consistent.

Minor typos:

On page 15, line 293, there is a capital ‘I’ used for ‘in’, which should be lower case.

On page 15 line 298, there should be a gap between ‘)’ and ‘effectiveness’.

On page 15 line 303, the sentence starts ‘All several sensitivity analyses’, either ‘all’ or ‘several’ can be deleted here.

Page 17, line 337, part of the sentence reads: ‘such as assessment of sweat, to gain….’, I was not sure what this meant or referred to.

Page 18, line 358, ‘e.g.’ is included where it isn’t needed.

Reviewer #3: The authors propose an economic evaluation of message frame-tailoring and content-tailoring on smoking cessation behavior. It is not clear from the existing structure of the manuscript as to why this problem is important and what is the related literature and related gap. The paper can be structured better in terms of motivating the readers as well as discussing about terms such as economic evaluation, message framing, and content framing. There are multiple ambiguities about the study design and choice of constructs/variables such as how is message-framing from content-framing and why was smoking abstinence over 7-days considered as the CEA outcome and quality of life considered as CUA outcome. Finally, though the title reads computer-tailored, the digital health aspect is either missing or not clear in this study. That is, the study could have been conducted using messages and contents printed on paper-surveys as medium. 

I would recommend the paper to be rejected in its current form as it does not appear to align with mission and scope of the journal, does not have significant results, and lacks clarity in current form.

6. PLOS authors have the option to publish the peer review history of their article (what does this mean?). If published, this will include your full peer review and any attached files.

**Do you want your identity to be public for this peer review?** For information about this choice, including consent withdrawal, please see our Privacy Policy.

Reviewer #1: Yes: Nicola McMeekin

Reviewer #3: No

---

## [Editor Report · Decision Letter 1]

22 Jul 2022

An economic evaluation of an online computer-tailored smoking cessation intervention that includes message frame-tailoring : A randomized controlled trial

PDIG-D-21-00116R1

Dear Dr Altendorf,

We are pleased to inform you that your manuscript 'An economic evaluation of an online computer-tailored smoking cessation intervention that includes message frame-tailoring : A randomized controlled trial' has been provisionally accepted for publication in PLOS Digital Health.

Best regards,

Valentina Lichtner

Academic Editor

PLOS Digital Health

Note: There is a typo in a revised section, where it says :

'cortisol in the salvia'

As you explained in your response to reviewers' comments, I believe you mean 'cortisol in the saliva'